# Enhancing Interprofessional Collaboration in Perioperative Setting from the Qualitative Perspectives of Physicians and Nurses

**DOI:** 10.3390/ijerph182010775

**Published:** 2021-10-14

**Authors:** Amalia Sillero Sillero, Neus Buil

**Affiliations:** 1Nursing School of Mar (ESimar), University of Pompeu Fabra, 08003 Barcelona, Spain; 2Department of Perioperative Nursing, Hospital de la Santa Creu i Sant Pau, 08025 Barcelona, Spain; nbuil@santpau.cat

**Keywords:** interprofessional collaboration, nurse–physician relationships, communication, teamwork, surgical team, shared decision making, safety patient

## Abstract

Communication failures were a leading cause of sentinel events in the operation room due to frequently the communication breakdown occurs between physicians and nurses. This study explored the perspectives of surgical teams (nurses, physicians, and anaesthesiologists) on interprofessional collaboration and improvement strategies. A surgical team comprising eight perioperative nurses, four surgeons, and four anaesthesiologists from a university-affiliated hospital participated in this qualitative and phenomenological research from December 2018 to April 2019. Data were collected in in-depth interviews and were used in a thematic analysis according to Colaizzi to extract themes and categorised codes with the ATLAS.ti software. The result is presented in three generic categories: Barrier-like disruptive behaviours and lack of coordination of care; consequences by safety threats to the patient; overcoming barriers by shared decision making among professionals, flattened hierarchies, and teamwork/communication training. The conclusion is that different teams’ perspectives can facilitate genuine reflection, discussion, and implementation of targeted interventions to improve operating room interprofessional collaboration and overcome barriers and their consequences. Currently, there is a need to change towards interprofessional collaboration for optimal patient outcomes and to ensure all professionals’ expectations are met.

## 1. Introduction

Communication failures were a leading cause of sentinel events in the operation room due to frequent communication breakdown that occurs between physicians and nurses [1,2]. Improving interprofessional collaboration (IPC) can be an essential factor for successfully meeting quality and safety goals in care settings [3,4]. IPC is the coming together of various health and social care professionals who solve problems and ensure patients’ health outcomes through effective communication, responsibility, and clinical competence. Additionally, trust, mutual respect, and the recognition of complementary knowledge develop over time [3,5]. Various studies suggest the need for collaboration between healthcare professionals to provide care [6,7]. IPC has improved quality and patient safety in various hospital settings [8]. Health professionals in the surgical environment have a common bond of concern for excellence in patient safety and must ensure its achievement [9]. Although the perioperative setting requires high-level surgical teams, few studies on interprofessional collaboration have been conducted due to the complexity of patient needs and the surgical process [6]. There is insufficient understanding regarding interactions between surgeons, anaesthesiologists, and perioperative nurses.

IPC has been studied in some systematic reviews proposing to compile research evidence to determine its effects and contributions. These reviews only included randomised trial (RCT) designs and quantitative research outcomes, but there is insufficient evidence from intervention studies to reveal clear conclusions [7].

However, the results of several systematic reviews of IPC interventions suggested their effects may vary according to the healthcare setting [10]; this variable is always standardised or controlled in these designs.

The study of context is better through qualitative and mixed-methods research designs because such methods provide more significant insights into the conditions under which IPC works. [11]. Thus, qualitative studies in different settings care are increasing [12]. 

Over the last years, the relationship between interprofessional teamwork and patient safety, as well as roles, performance, and communication of the surgical team communication, has gained increasing attention [13,14,15]. Several kinds of interprofessional practices have been used in different studies, such as checklists and training teams [7]. Therefore, this qualitative study aimed to understand IPC in that setting. We limited our research to the operation room, as these care settings require collaborative teams of different professionals. 

Therefore, the outcomes of this study can provide information on these challenges and present strategies and theories. Moreover, understanding the communication methods within an organisation from the health professional’s perspective can help leaders identify problems and support the development and implementation of prevention strategies [16]. Thus, the study explored the views of surgical teams (perioperative nurses, surgeons, and anaesthesiologists) on interprofessional collaboration and improvement strategies. Our research question was, “what are the surgical team’s experiences and perspectives on interprofessional collaboration?”

## 2. Materials and Methods

### 2.1. Aim

The study explored the views of surgical teams (nurses, physicians, and anaesthesiologists) on interprofessional collaboration and improvement strategies. 

### 2.2. Design

A qualitative descriptive phenomenological study was used to understand the lived experiences and views of the surgical team about the difficulties of teamwork. The consolidated criteria for reporting on qualitative research (COREQ) were followed [17]. The data collection method was semi-structured interviews conducted from December 2018 to April 2019.

### 2.3. Setting and Sample

The sample selection was based on purposeful sampling among surgical team members to the same surgical area in Spain. The inclusion criteria were surgical nurses, surgeons, and anaesthesiologists working in a perioperative setting having worked in the area for at least one year; voluntary participation in the study, and speaking Spanish. A snowball sampling strategy was followed to identify key informants from the interviews that were conducted. A total of 16 participants have interviewed: eight nurses, four anaesthesiologists, and four surgeons. 

### 2.4. Data Collection

The sample size was by the saturation of information. Data were collected via face-to-face in-depth interviews. The participants were interviewed using a semi-structured guide that we developed in the context of the relevant literature of IPC and contained the questions listed in Table 1 [18,19].

Moreover, we conducted the interviews in a private setting (meeting room) inside the hospital during the daytime working hours. The interview form included 12 questions related to interprofessional collaboration in surgical teams. Individual in-depth interviews lasted approximately 35 min (ranging from 35 to 50 min). We recorded the interviews with the participants’ written permission (informed consent), guaranteeing their confidentiality and anonymity.

The interviews were transcribed, coded, and examined for themes with the help of ATLAS.ti computer-assisted qualitative data analysis software, which provided tools to help code, organise, and analyse the large amount of raw data collected. We used Colaizzi’s [20] phenomenological research method for an in-depth understanding of the surgical team’s experience of IPC. 

The phenomenological method is an inductive method for identifying the meaning and essential structure of subjective experience through statements. 

The researcher accepts and understands the subjective perspective of the participants without any prejudice.

It consists of the following seven steps:Familiarisation with the interview by listening to it several times;Extraction of words and meaningful sentences—first, we highlighted the significant statements from the transcript;Formulation of meanings—in this step, we used codes as key concepts independently and noted the extracted topics;Organisation of themes into clusters;Description exhaustive—we defined the relation of each theme to the phenomenon;Structuring the topics to define significant subtopics or those similar to each other, leading us to a consensus;Verification—the resulting data were shown to the participants to review and verify if they agreed with the topics and subtopics extracted.

### 2.5. Data Analysis

Content analysis of the interviews was conducted in the following process based on different stages: the first stage was the transcription and subsequent immersion in the discourses, reading the phrases, and the notes collected in the field diary.

In the second stage, units of analysis of the interviews were defined and coded for emerging categories. Next, the results were structured, and their meaning and relationships were investigated to understand the phenomenon under study. An inductive analysis enabled themes and added new properties and attributes to the categories with new participants.

The research was guaranteed by triangulation for data accuracy and reliability [21]. Two expert nurses who were not perioperative but experienced qualitative research and did not participate in this study reviewed the data analysis. The quality of this research was conducted using a checklist [17]. Field notes extracted and analysed from the interviews were also used, and always check the data with the participants [22]. This research follower the criteria were ontological and authenticity [23].

## 3. Results

### 3.1. Demographic Profile

Sixteen participants were recruited: eight nurses, four anaesthesiologists, and four surgeons specialising in cardiac surgery, and none refused participation.

All nurses interviewed were female, and in the other disciplines, there were equal numbers of male and female participants. Ages ranged from 24 to 60 years old, the average age was 42 years (SD = 11.2), and years of experience averaged 14.1 years (SD = 12.1); see Table 2.

### 3.2. Thematic Findings

We compiled an extensive amount of information from the 16 interviews conducted.

We summarised findings in fifteen themes, seven theme clusters, and three generic categories: (a) barriers/challenges, (b) consequences/effects, and (c) suggestions/overcoming barriers.

The study participants indicated barriers, reflected on consequences to IPC, and made precise suggestions, listed in Table 3.

#### 3.2.1. Category 1: Barriers/Challenges

The following were identified as causes or problems of perioperative communication: organisational, individual, and perioperative environment factors.

The problems shown were mainly the bed number or the medical diagnosis to identify the patient. For example, a wrong patient transfer can be made if team members use the name of the surgery or the bed number to refer to the patient instead of the patient’s name.

Nurse 1 stated, ‘bypass patient’s [blood] results …. There are various bypass patients in the ward …. The same goes for the operating room; Is the second cardio surgery ready? … which patient is it? What is your name?’

Nurse 1 also explained, ‘Doctors sometimes do not write antibiotic prescriptions; I sometimes the physicians confuse nurses. They also do not change the orders if there are changes in treatment. We need good communication and adequate institutional protocols’.

Additionally, role misunderstandings exist; for instance, nurses assume they have to solve administrative tasks, such as reviewing the patient’s informed consent and reports from other consulting services, and sometimes feel that these problems do not concern them. These are frequent and prominent occurrences. 

Nurse 2: ‘We are to do everything, e.g., the patient can’t go to surgery without signing their consent. I do not know any nurse out there who refuses; I have not seen any. Some surgeons might hold a nurse responsible if you refuse. If you refuse, someone else in your team will do it for you. It is stressful’.

Nurse 3: ‘Patients must come with informed consent. We are warned that we have to review it. If we do not do so, we have a problem. Sometimes I’m very concerned and angry and I tell them’.

Anaesthesiologist 1: ‘There are frequently changing responsibilities, I should be able to change the position of the patient in bed without permission of the surgeon, and I do it, but the other nurses are avoiding [it] because they are afraid that the surgeon will be angry’.

Another factor was inadequate time for preoperative preparation of patients and the operating room.

Nurse 7: ‘There is little time to prepare because there are few of us to do the job, to have everything ready. We need to prepare the patient and documents and sometimes something can be an accessory left …. This leads to the confrontation between theatre staff and the nurse in the ward’.

Nurse 1: ‘There is no time …. I can’t report and prepare everything, and I need someone to help me …. I can even send the patient to another operation’.

Inadequate time for preoperative patient preparation was also expressed. The stress caused by the accelerated pace and pressure to complete the activity can decrease communication quality, as explained by the following participants:

Surgeon 3: ‘I cannot work on the patient the way I want, and it affects my work’.

Anaesthesiologist 2: ‘Of course, sometimes you feel understaffed’.

Surgeon 1: ‘The operating room is like this … everything goes fast, practically everything must be done immediately, without waiting, with tension. I’m the responsible’.

Regarding individual factors for perioperative communication failure, especially related to physicians’ disruptive behaviour and poor work dynamics, the participants explained these behaviours as follows:

Nurse 9: ‘They dare to tell us that we are slow if the patient has not gone to the operating room, and in reality, they have not even arrived on time …. and they want the patient to be ready …’.

Nurse 1: ‘In fact, there is no stressful environment in the room, but … when the surgeon has a problem, he screams for ten minutes and leaves’.

These types of dynamics were considered inappropriate. 

Nurse 8: ‘I am concerned; the surgeon sees us as an assistant, not as a co-worker’.

Anaesthesiologist 4: ‘I am uncomfortable; even if we have a good relationship, the nurses have a perception of subordination’.

#### 3.2.2. Category 2: Effects/Consequences

The consequences or effects were decreased staff retention, avoidance of team members, and threats to patient safety. Team disruptions and the high levels of stress of the professionals caused these effects. These problems led to evasion among some of the members, and even leaving the job.

Anaesthesiologist 2: ‘Sometimes there are problems with a certain surgeon, and you do not want to work with him or prefer not to be near that person’.

Surgeon 2: ‘It is a pity, but sometimes some people make nurses leave the perioperative setting’.

Surgeon 3: ‘We are short of time and thus cannot dedicate enough time to them (team). However, I know communication is essential, and a care plan has to be laid out to nursing staff, colleagues and aides who are participating in care’.

Some of the participants showed that the effects of communication failures were inadequate teamwork, loss of effective communication, and therefore, threats to patient safety.

Nurse 4: ‘they do not wait, there might be a confusion, you do not go fast in finishing, the count is missing, there is one missing, I think, I wait’.

Nurse 3: ‘The last month I report a medical error associated with verbal order. I believe messages delivered via verbal orders are jeopardise the safety of the patient’.

Nurse 5: ‘Why don’t you listen to me …. I am going to indicate that the count is not done it is your responsibility I will indicate that I have already signed’.

#### 3.2.3. Category 3: Suggestions/Overcoming Barriers

Effective communication needs support from organisational protocols. Operating room briefings are an opportunity to share surgical teams’ mental models and improve patient safety; procedures may be established that ensure that surgical teams are available when critical decisions are made. There were parallels between the barriers of communication and suggestions to prevent them among the categories extracted. Recommendation themes included creating clear job descriptions, institutional security protocols, team spirit building, and strengthening social interaction between different team members. Two categories were established to create organisational norms in the security protocol and team mental model sharing.

The participants reported that other team members do not occasionally respect their work, which caused stress for team members.

Anaesthesiologist 3: ‘When working as a team, responsibilities are shared, but sometimes they, the surgeons do not consider it. Therefore, it is essentially used the checklist furthermore the briefing of the surgical team’.

Strategies to empower professionals mainly included rounds and meetings to inform each other, as well as to create team spirit and a positive and healthy work environment. Participants explained this as follows:

Nurse 1: ‘The work of our team is excellent, there are no problems or possible mistakes, all this creates an atmosphere that is positive, we already know each other so much…’.

Nurse 2: ‘Above all, it is necessary to instil a team spirit, share responsibilities… and consider everyone when making interventions. I think that the pre-intervention information rounds are perfect for me when they explain to you what we are going to do and need’.

Nurse 7: ‘Even if we are competent and excellent, always I tell them we need help and work together’.

Anaesthesiologist 3: ‘Everyone participates using a closed-loop … and maybe we needed others safety tools to believe checklist implementation allows teams to focus on the patient’.

Nurse 3: ‘It is fundamental to have a healthy work environment without stress so that we can interact and work better, know each other, and talk about what we are going to use at the beginning of the operation’.

Surgeon 1: ‘Attempts are doing very good regarding teamwork communication, that any decisions are team decisions. Moreover, always we have must good and communication’.

Nurse 6: ‘When we make the checklist, we communicate consciously, but what I like best is that each of us assumes our responsibility’.

Participants explained that the surgical team needs to be trained on problem-solving methods, stress management, time management, and team empowerment to prevent communication failures. Some members worked towards patients’ care needs, trying to avoid a more traditional hierarchical model.

Surgeon 4: ‘You have to take one step towards one another; I always had good experiences in doing so promote effective communication’.

Nurse 3: ‘It would be advisable that they teach us how to manage the tension and stress with the team; I cannot control myself sometimes; I am very impulsive’.

Nurse 4: ‘When we worked joined, we together improved. We work to close the loop, and I am sure of safety patient’.

Another suggestion from the participants was the inclusion to improve social interaction among all team members.

Anaesthesiologist 4: ‘Activities can be done outside the operating room. When prejudices are removed, communication becomes smoother. I can approach the one and express myself more appropriately’.

## 4. Discussion

In this study, participating surgeons, anaesthesiologists, and nurses explained their experiences in interviews about the importance of interprofessional communication and collaboration for surgical patient safety, its principal barriers, and the strategies and techniques for better communication quality in the operating room.

Some weaknesses of perioperative care with potential for improvements were identified. The participants explained numerous communication patterns that can create misunderstandings or incomplete information [24]. Most nurses indicated they wanted a more frequent collaboration with physicians and told that failures in teamwork and communication were essential sources of mistakes [25,26]. Considered barriers among participants included excessive workload, inadequate time for preoperative patient preparation, and non-compliance with the procedures for communication and the surgical checklist. In addition, participants often performed tasks that they should not, increasing their workload [27,28].

Other themes that emerged were lack of personnel and lack of motivation. Participants explained that performing tasks outside their practice, combined with inadequate staffing, leads to communication failures. A substantial relationship exists between medical errors, patients’ safety, and quantity of personnel [2]; thus, it can improve by allocating adequate human resources. In this study, participants indicated that some of the communication problems are related to stress caused by the physical environment of the perioperative areas [29]. In addition, prolonged operations and delayed or non-existent breaks can create tension among staff members and make communication difficult [30,31]. Some team members perceived the necessity of interprofessional interactions and recognition of each other’s role. Although some findings indicate that the nurses and physicians in this study wished interprofessional collaboration in the operation room, the participants described limitations due to individual and organisational factors. The relationship between physician and nursing leadership silos may create obstacles to optimal teamwork and accountability. According to participants, the differences between nurses and doctors are due to hierarchies and responsibilities due to socio-cultural background, gender, beliefs, and individual attitudes. The literature supports those findings [32,33]. Additionally, the perioperative nurses identified that increased stress and adverse situations with surgeons had caused some nurses to leave the hospital [34]. The surgeon traditionally has the perioperative team leader role, and they perceive the workplace more positively than others. However, other team members could be the leaders. Recent articles have addressed the role of anaesthesiologists, who are ideally positioned to bring expertise as perioperative physicians because of their understanding and ability to assess, evaluate, and prepare patients with a multitude of complex comorbidities for their procedure and their ability to manage these complex comorbidities intraoperatively and postoperatively [35,36]. The literature shows that interprofessional education can define roles and responsibilities among interprofessional teams [37].

Several participants in this study reported that communication problems had caused a wide range of issues, from leaving the profession to reduced patient satisfaction. However, the most significant consequence is the threat to patient safety. Medical errors, especially in the perioperative environment (foreign body retention, wrong-site surgery, improper operation, or death), are the most severe adverse hospital events. When millions of procedures were reviewed, it was found that a large proportion of this harm is preventable by eliminating deficiencies in teamwork (absence of coordination, poor communication, and lack of a shared mental model) [1]. There is a significant problem; therefore, the interprofessional collaboration of surgical teams is necessary to maintain and improve patient safety [38]. Participants reported that clear role descriptions and establishing team meetings and social activities would be essential in preventing adverse events [28,39].

According to the literature, over a decade ago, few or no changes have occurred in clinical practice, and surgical teams still must deliver quality surgical care without adequate resources. There are still many limitations in interprofessional collaboration.

Our research showed that leaders only partially perform their duties about establishing teamwork in hospitals, leaving employees with few opportunities for personal involvement and that nursing leaders allow the subordination by physicians. From the viewpoint of organisational culture, this result poses a challenge for managers and leaders in nursing and medicine, who can change the current hierarchy trend by promoting teamwork culture, innovation, integration, and personal involvement of all health care professions. Only by encouraging these values will the hospital work environment be a success.

Additional research is needed, but interprofessional collaboration and teamwork will not prosper without changes in the organisational culture [39].

Effective and assertive communication with colleagues, reflecting on practice, incident reporting, and redefining roles are some of the strategies that help surgical teams overcome the barriers and learn how to engage in more team-based work through briefing, surgical checklists, and engaging in closed-loop communication [26,40].

### Strengths, Limitations, and Future Directions

In addition to its practical implications, this study contains numerous strengths. First, all the professionals involved in the surgical team revealed different opinions that would have been missed with equal professional participants. Second, the qualitative approach allowed knowing the phenomenon, uncovering the importance of the surgical team interprofessional collaboration. Finally, the researchers involved in this study explained and broadened perspectives during the analyses of the content of the interviews [11].

Further research is required to confirm our outcomes and compare them with the results obtained from interviews in other perioperative settings or other care settings. After this study, an appropriate amount of time should be given so as to implement strategies for facilitating interprofessional collaboration. There are factors with regard to the organisational culture of the operation room that represent potential barriers and should always be identified. Moreover, should strategies for skills training of teams be assessed in prospective studies to demonstrate whether they can improve the interprofessional collaboration during the surgical process and the long-term well-being of the surgical team?

This study was carried out in only one hospital setting and may not represent other hospitals or departments, although we reached saturation regarding the themes raised during the interviews after interviewing 16 participants. Moreover, most participants volunteered for the study, which may have biased the content of their statements. Finally, the male:female ratio in our sample was representative of our eligible physician population; however, nurses were all female. Therefore, it may have limited the gender representation among the participants.

## 5. Conclusions

Overall, the results show that different perspectives of the surgical team about inter-professional collaboration can facilitate genuine reflection, discussion, and implementation of surgical care improvements. Participants’ perceptions identified the need for high interprofessional collaboration in the operation room, rather than simple coordination and cooperation. There are many barriers at the team, organisational, hierarchical, environmental, and individual levels. The surgical teams require a change, and they must overcome the obstacles. An interprofessional practice involves learning how to dialogue more effectively and assertively with colleagues, reflecting on practice, such as discussing team-based work through briefing, and engaging in redefining roles that lead to surgical checklists, engaging in closed-loop communication, and addressing issues through incident reporting. In response, researchers need to communicate with practitioners and determine how to help the latter develop an exploratory folk theory with explanatory theory founded on evidence.

The findings of this study possibly present the strategies that aim to improve interprofessional collaboration of healthcare providers in surgical care. The managers and administrators could produce efficient teams in perioperative settings or other care settings with optimal patient outcomes whilst meeting productivity targets and ensuring all professional expectations are met. This is a required condition.

## Figures and Tables

**Table 1 ijerph-18-10775-t001:** Questions participants’ interview guide (IPC).

	Questions IPC
1	What is your experience of IPC in your team?
2	What do you think promotes IPC?
3	What do you think inhibits IPC?
4	Does the perioperative environment encourage positive feedback? How?
5	What significance do you think IPC has in patient safety?
6	What is your experience of communication between the surgical team professions?
7	What do you think are important issues to report regarding patients with clinical deterioration?
8	Can you give concrete examples of how your professional knowledge is important in your IPC?
9	What is your experience of sharing different views across the professions?
10	How does the management in your department facilitate IPC?
11	What is needed for IPC to develop in a good way?
12	What is your knowledge about the techniques for better achievement for the surgical team?

**Table 2 ijerph-18-10775-t002:** Characteristics of study participants.

Participants	Sex	Discipline	Age	Experience
1	M	Surgeon 1	50	24
2	F	Surgeon 2	48	25
3	M	Surgeon 3	30	6
4	F	Surgeon 4	28	4
5	F	Perioperative nurse 1	44	22
6	F	Perioperative nurse 2	25	6
7	F	Perioperative nurse 3	24	1
8	F	Perioperative nurse 4	46	25
9	F	Perioperative nurse 5	33	12
10	F	Perioperative nurse 6	60	30
11	F	Perioperative nurse 7	32	10
12	F	Perioperative nurse 8	36	18
13	F	Anaesthesiologist 1	35	10
14	M	Anaesthesiologist 2	42	15
15	F	Anaesthesiologist 3	27	4
16	M	Anaesthesiologist 4	29	6

**Table 3 ijerph-18-10775-t003:** Categories, theme clusters, and themes.

**Categories**	**Theme Clusters**	**Themes**
Barriers/challenges	Organisational factors	Carrying out administrative tasks
		Fast and intense environment
		Lack of personnel
		Work overload
	Psychological factors	Tensions inside the surgical team
		Disruptive behaviours
	Educational factors	Professional hierarchies
Consequences	Low nurse retention	Intention to abandon
	Problems with the team and Safety threats	Non-communication/verbal abuse
		Adverse events
Suggestions/ overcoming barriers	Compliance with safety protocols.	Standard procedures/checklist
	Sharing practice among professionals	Regulations of job
		Round or Briefings
		Favourable work environment

## Data Availability

The data presented in this study are available on request from the corresponding author.

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
