# Peer review of "Enhancing Interprofessional Collaboration in Perioperative Setting from the Qualitative Perspectives of Physicians and Nurses"

_ijerph, 2021, doi:10.3390/ijerph182010775_

Round 1
Reviewer 1 Report
Reviewer: Dear authors, I read the article entitled: Enhancing interprofessional collaboration in perioperative settings from a qualitative perspective of physicians and nurses [change a little bit]. The article main message is to enhancing interprofessional collaboration in the perioperative setting. This question merit consideration. The manuscript is writing sufficiently. If the journal accepts this type of manuscript below, you can find my main suggestion as Anesthesiology. Anesthesiologists are uniquely positioned to serve as "perioperativists" because of their understanding and ability to assess, evaluate, and prepare patients with many complex comorbidities for their procedure and their ability to manage these complex comorbidities intraoperatively and postoperatively. This global understanding will allow anesthesiologists to drive the standardization of care needed to reduce risk and optimize perioperative outcomes.** **Reference: The Perioperative Surgical Home: How Anesthesiology Can Collaboratively Achieve and Leverage the Triple Aim in Health Care Thomas R. Vetter, MD, MPH, Arthur M. Boudreaux, MD, Keith A. Jones, MD, James M. Hunter Jr, MD, and Jean-Francois Pittet, MD Anest Analg 2014;118:1131-6 Della Rocca G, et al. Preoperative Evaluation of Patients Undergoing Lung Resection Surgery: Defining the Role of the Anesthesiologist on a Multidisciplinary Team. J Cardiothorac Vasc Anesth. 2016 Apr;30(2):530-8. doi: 10.1053/j.jvca.2015.11.018. Epub 2015 Dec 1. PMID: 27013123. The PSH requires a physician team leader, the "perioperative," who provides seamless continuity of current best care practices while actively involving the patient, family, and the other health care stakeholders and providers, including the primary care physician. Several clinical specialists could serve as this perioperativist in the PSH. The surgeon has traditionally served as the perioperative team leader. However, .........................., Anesthesiologists are uniquely positioned to serve as perioperativists because of their understanding and ability to assess, evaluate, and prepare patients with a multitude of complex comorbidities for their procedure and their ability to manage these complex comorbidities intraoperatively and postoperatively...........(**) Fix this article so that anesthesiologists can point to it when they hand a copy to their colleague surgeons, cards folks, pulmonologists, etc. While I have a pretty good relationship with these folks, I am sure that they have little knowledge of the latest literature outside of their field. I am damn sure that very few of them have any clue about anesthesia, except that we delay cases and are responsible for everything that goes wrong in the theatre. 1. Abstract: ok 2. Introduction: ok 3. Methods: ok 4. Result: ok 5. Discussion: Please discuss my previous comment above. 6. Limitation: ok 7. Conclusions: Ok 8. Tables: ok 9. References: to be improved Best Regards
Reviewer 2 Report
This study is a qualitative data analysis interpreting qualitative research data in an interprofessional collaboration from the perspective of physicians and nurses. Meaningful information was identified and organized it into themes or categories.
-
- The Colaizzi's Method follows seven data analysis steps. Could you please define this?
- I suggest adding a completed COREQ list for the reporting of your qualitative research process
- Table 3: could you please explain "low nurse retention"
- To get the exact meaning of the statements, the translation of the Spanish interview segments (line 152 to line 275) should be revised and translated by an native English speaking person
- in which surgical discipline is the team working?
- The sentence line 310-315 is written twice
- CIRS: do you have a critical incident reporting system in your clinic? Some of your recurring issues might have been solved with the implementation of such a system.
- One of the major flaws of the study is that you report on one group and there is no comparator in another hospital or other system. So, you mention facts of your OR organisation that represent barriers. Which barriers do you mean and how might you overcome those?
- line 257: surgeon 1 says that "attempts are doing very good regarding teamwork communication. That any decisions are team decisions. Moreover, always we have must good and communication" This is conflicting to most other statements. What do you deduce out of that statement?
- Could you expand on the question 11) what is needed for IPC to develop in a good way?
- Based on your own data, please expand on your statement that interprofessional collaboration and teamwork will not prosper without changes in the organisational culture (line330)
Round 2
Reviewer 2 Report
I fully agree with their prompt answers and explanations. I have nor further comments and agree with the actual revision of the authors.